# Research on the Simulation Model of Continuous Fiber-Reinforced Composites Printing Track

**DOI:** 10.3390/polym14132730

**Published:** 2022-07-03

**Authors:** Yesong Wang, Jiang Liu, Yipeng Yu, Qing Zhang, Hongfu Li, Guokun Shi

**Affiliations:** 1School of Mechanical Engineering, University of Science and Technology Beijing, Beijing 100083, China; wys_1993@126.com (Y.W.); g20198611@xs.ustb.edu.cn (Y.Y.); s20190513@xs.ustb.edu.cn (Q.Z.); m202120577@xs.ustb.edu.cn (G.S.); 2School of Mechanical Engineering, Jiangsu University of Science and Technology, Zhenjiang 212003, China; 3School of Materials Science and Engineering, University of Science and Technology Beijing, Beijing 100083, China; lihongfu@ustb.edu.cn

**Keywords:** additive manufacturing, CFRTPC, fiber track, line-following model, minimum curvature, track simulation

## Abstract

The rapid development of additive manufacturing technology (AM) is revolutionizing the traditional continuous fiber-reinforced polymer (CFRP) manufacturing process. The combination of FDM technology and CFRP technology gave birth to continuous fiber reinforced thermoplastic composites (CFRTPC) 3D printing technology. Parts with complex structure and excellent performance can be fabricated by this technology. However, the current research on CFRTPC printing mainly focuses on printing equipment, materials, and the improvement of mechanical properties. In this paper, the CFRTPC 3D printing track errors are investigated during the printing process, and it is found that the polytetrafluoroetylene (PTFE) tube in the nozzle of the printer head is often blocked. Through detailed analysis, a line-following mathematical model reflecting the deviations of the CFRTPC printing track is established. According to the characteristics of the fiber and its track during actual laying, a modified line-following model, without the minimum curvature point, is further proposed. Based on this model, the actual printing track for the theoretical path is simulated, the process tests are carried out on the printing track at different corner angles, and the relevant rules between the parameters of the model and different corner angles are obtained. The mathematical model is verified by experiments, and the clogging problem of the printer head caused by the fiber track error is solved, which provides theoretical support for the rational design of the fiber track in CFRTPC printing.

## 1. Introduction

3D printing [1] is an additive method for rapidly manufacturing products, compared with traditional “subtractive manufacturing”. It is another technological revolution in the field of manufacturing, which shows the potential and vitality of personalized creation in the new era, and is widely used in mold manufacturing, aerospace, automotive, home appliances, medicine, and other fields. Among the many 3D printing technologies, fused deposition manufacturing (FDM) attracts a lot of attention due to its low cost and convenient operation. FDM continuously extrudes layers of thermoplastic polymer materials (such as PLA, ABS, PA, etc.) to produce complex parts. However, due to the limitation of the properties of polymer materials [2], the parts prepared by FDM have some defects, such as low strength, inability to withstand excessive loads, and low molding accuracy, which cannot meet the needs of industry. Continuous fiber can effectively improve the mechanical properties of different matrix materials in 3D printing, such as thermosetting resin materials, thermoplastic resin materials, and concrete [3,4]. Parts made with continuous fiber-reinforced polymer (CFRP) composites have excellent mechanical properties, including high specific stiffness and strength [5], and receive extensive attention from the industries of civil engineering and many other fields [6,7,8]. Recent studies have shown that CFRP composites have great potential for developing components with versatile [9,10] properties. However, the current CFRP processing technology still faces many challenges, among which the biggest challenge [11] is how to efficiently combine CFRP materials with matrix resin at low cost so that it has good consolidation and fiber orientation [12]. The current general CFRP molding process is divided into two stages; first, the CFRP composites are laid into the molds, then they are subject to heating, pressing, thermal insulation, and curing. Because this method requires special molds, the cost of the molds are too high [13] for a single part. The use of special devices for the heating, pressing, and curing process increases the manufacturing cost too. Therefore, CFRP composite manufacturing is not suitable for small batch production or individual manufacturing of the parts.

The combination of FDM and continuous fiber-reinforced thermoplastic composites (CFRTPC) gave birth to CFRTPC 3D printing technology, which simplified CFRP manufacturing with low cost and high automation. Widely considered by scholars and industry [14,15], this method is mainly used for the low to medium volume and customized parts, and can realize rapid prototyping of parts with complex geometries. Both CFRTPC printing and general FDM are essentially stacking a series of discrete layer planes to fabricate parts. Unlike ordinary FDM, when CFRTPC is printed, CFRTPC needs to be continuously and accurately laid on the surface of the substrate according to the length of the planning path. The current CFRTPC printing is mainly divided into two types: the two-step method (independent extrusion method) and one-step method (co-extrusion). The first step of the two-step method [16] is to produce CFRTPC prepreg [17]. In this process, a screw extruder is used to provide a large pressure for the molten polymer so that the dry fiber can be fully infiltrated by the thermoplastic material. The second step is to feed the CFRTPC prepreg filaments into the 3D printer nozzle for printing. The one-step method [18] is to feed continuous dry fibers and thermoplastic polymers into the nozzles through the two inlets of the printer head, respectively, so as to immediately print after infiltration. Since CFRTPC 3D printing can accurately lay the CFRTPC track of each layer, designers and manufacturers can design and optimize the CFRTPC track for each layer, thereby improving design freedom and mechanical performance.

Research on CFRTPC 3D printing is still in its infancy, and several companies in the market developed CFRTPC commercial one-step and two-step 3D printers and supporting software. Based on the two-step method, Mark One and Mark Two series printers were developed by Markforged [19], which use continuous carbon fiber-reinforced nylon to print parts with mechanical properties that are an order of magnitude higher than those printed by ordinary FDM printers; this opens up new application scenarios in the personal manufacturing market and the manufacturing of industrial lightweight components. The Composer A3 and A4 series printers developed by Anisoprint [20] use composite fiber co-extrusion (CFC-composite fiber co-extrusion) technology, and their supporting software can change the fiber direction, volume ratio, and density. The Aqua 2 printer developed by AREVO [21] can print large-area continuous carbon fiber parts, and it is successfully applied to the manufacture of bicycle frames. Switzerland 9T labs [22] added a dedicated continuous fiber printing head to a general FDM printer, and developed the continuous fiber laying software fibrify, which can optimize the fiber laying direction according to the load of the part and realize the production of composite structural parts with complex internal fiber orientation and the internal porosity controlled below 2%. Arris Composites [23] used its innovative additive molding manufacturing process to manufacture a carbon fiber truss that nearly doubles the specific stiffness of I-beams, and this manufacturing process can achieve 100% recycling. Based on the principle of the one-step method, the COMBOT-1 printer, jointly developed by Shanxi Feibo Technology Company and Xi’an Jiaotong University [24], can realize the rapid manufacture of complex structures. The printer uses the prepared CF/PLA material to print honeycombs. The results show that adding continuous fibers only increases the mass by 6%, and the compressive stiffness and energy absorption of the honeycomb structure increase by 86.3% and 100%, respectively.

For the improvement on mechanical properties, J.M. Chacón et al. [25] studied the effects of fiber laying direction, fiber volume fraction, and other process parameters on the mechanical properties of CFRTPC samples, as well as compared the effects of carbon fiber, glass fiber, and aramid fiber on PA reinforcement. Wang et al. [26] studied the mechanical effects of process parameters, such as printing temperature, speed, layer height, and fiber volume fraction, on the standard mechanical sample. When the fiber volume fraction was 5.21% and 6.24%, tensile and flexural strengths were increased fourfold and twofold, respectively. Dong [27] studied the reinforcement effect of different continuous fibers on nylon matrix materials and found that the tensile stress of carbon fibers was 165 MPa higher than that of glass fibers and aramid fibers through mechanical properties tests. Khalid Saeed et al. [28] studied the mechanical properties of nylon specimens with continuous fibers using Mark Two, and compared them with the nylon specimens without reinforcements; the elastic modulus and tensile strength were increased by 603.43 MPa and 85 GPa, respectively. Guang Liu et al. [29] studied the future application directions of the additive manufacturing of CFRTPCs. From the perspective of functional requirements, they introduced some new applications of CFRTPC 3D printing in shape deformation, sensing, and energy storage. This literature shows that the addition of CFRTPC can greatly improve the mechanical properties of printed parts, and the application of this technology is still expanding. However, due to the internal defects of the printed parts and the low proportion of fiber volume fraction, the mechanical properties of CFRTPC 3D printing parts cannot be compared with those made by traditional CFRP methods [30], and the internal defects of the parts make it impossible to increase the fiber volume fraction proportion. Therefore, the biggest problem to be solved at present is why defects are formed into the printed parts and how to reduce them. On the one hand, the causes of internal defects involve the design and optimization of CFRTPC track. Based on the FDM layer-by-layer printing pure matrix, in the independent extrusion CFRTPC printing method, the printing layers containing CFRTPC are inserted among the pure matrix printing layers as the reinforcement layers. When one fiber track printing ends in a layer, the fiber nozzle needs the fiber to be cut and to go on printing the next fiber track or to be switched for the printing matrix. When there are multiple fiber tracks in a single layer, there will be multiple cutting and jumping steps, and too many fiber cutting points will increase the internal defects of the part.

Yiwen Tu et al. [31] studied the discontinuity of the printing path of the CFRTPC printing process. They assembled the cutting device into the nozzle and proposed a new front-end cutting algorithm, whose parameters are compensated by analysis and tests, and designed the CFRTPC concentric circle filling method; finally, a good effect was obtained for when the CFRTPC jumps between 3D printing layers. Yiming Huang et al. [32] developed a multi-scale strategy for CFRTPC that integrates the simultaneous optimization of fiber orientation and macrostructure topology by ingenious planning printing paths. Fuji Wang et al. [33] developed a novel CFRTPC track which can well avoid the jumps of continuous fibers, solve the dispersion problem of connecting paths, and reduce the number of cuttings in CFRTPC commercial equipment, but the method is only suitable to the simple parts of pure CFRTPC. Shengjie Zhao et al. [34] studied the influence of the CFRTPC angle and load direction on the mechanical properties of the specimen, and concluded that slight fiber dislocation has little effect on the structural stiffness of the part. 

The above scholars started from reducing cutting jump points, optimizing fiber track paths, reducing the number of turns, etc., which greatly reduced the internal defects of parts, but they did not consider the inconsistency between the theoretical track and the actual track of continuous fibers, especially at the corners of the track where a large misalignment deviation is. A few scholars found internal defects caused by the radius of the fiber curvature in their studies, but did not carry out systematic theoretical research. Therefore, it is of great significance to study the track error and its influencing factors. It will provide theoretical guidance for reducing the internal defects of the parts and rationally planning the fiber tracks. Since the printed fibers have a turning radius at the corners due to the tension of the continuous fibers themselves, the brittle CFRTPC cannot pass a smaller turn than the radius, otherwise, the continuous fibers will break. Even if fiber breakage does not occur at the corners, the actual printed track will leave a void area error from the design track. When printing complex shape parts, there are more error areas. The accumulation of error areas in the continuous fiber track increases the internal defects of the part, reduces the strength of the part, and also causes printing failures. Due to the printing track error, CFRTPC 3D printing cannot fabricate parts with high performance and complex shapes. 

In this paper, the causes of CFRTPC printing track errors are analyzed, and a mathematical model is established to simulate the actual printing path deviating from the theoretical path, and reflect on the errors between them. Additionally, process tests for the printing track at different corner angles are conducted. The correlation rules between the model parameters and different corner angles under the minimum error are investigated. The research in this paper can provide a theoretical basis for continuous fiber trajectory planning in the 3D printing of continuous fiber-reinforced composites. The optimized printing trajectory can effectively reduce the failure of nozzle blockage in the printing process and improve the printing reliability of CFRTPC. At the same time, the printing accuracy of continuous fiber can be improved, the internal porosity of 3D printed parts can be reduced, and the performance index of continuous fiber-reinforced composites can be further improved.

## 2. Materials and Methods

### 2.1. Printing Devices

In this study, the CFRTPC 3D printer BF300 jointly developed by Suzhou Bofei Yicheng Electromechanical Co., Ltd. and our laboratory is used, as shown in Figure 1a. The printing is based on the principle of a two-step method. The continuous fiber-reinforced thermoplastic composite filaments (CFRTPCFs) are pre-prepared before printing, and the matrix and (CFRTPCFs) are printed separately during printing. The printing principle is shown in Figure 1b. The printer adopts an innovative independent multi-nozzle structure design, one of which is an independent CFRTPCF printing nozzle, and the rest are nozzles for polymer (PLA, ABS, PA, etc.) printing, as shown in Figure 1c,d. The printer selects the different nozzles by the head change command (T). The equipment lays CFRTPCFs by the jointly developed software BFslicer 1.2, which can arbitrarily design CFRTPCF tracks within a single layer.

### 2.2. Printing Materials

The matrix used in this study is polylactic acid (PLA), which is provided by eSUN manufacturer [35]. The CFRTPC material used in this study is continuous glass fiber-reinforced filament /PLA (CGFRF/PLA), prepared by the device developed for preparation of CGFRF/PLA in the laboratory and then adopted for the best preparation process parameters of test [36]. The material parameters are shown in Table 1 and Table 2 below:

### 2.3. Printing Parameters

The software first slices the initial model into layers and then performs track planning within the layers. The track file (G-code file) is imported into the device for printing. The standard tensile specimen and the three-point bending specimen are shown in Figure 2, and the printing parameters are shown in Table 3.

## 3. Print Tracks

### 3.1. Analysis of Errors

In the actual printing test, there is a certain failure rate that occurs during the printing process. After dismantling the printer head, it is found that the PTFE (polytetrafluoroetylene) tube located in the nozzle, shown in Figure 3a, is often blocked by fibers, as shown in Figure 3b. Through many experiments, it is found that there are certain errors between the actual and planning fiber paths. The actual fiber paths are shorter than the planning paths, and the fiber extrusion length of each path is calculated according to the theoretical planning path. Some redundant fiber composites are not laid on the surface of the substrate and accumulate in the nozzle. As the printing tracks grow, especially as the turning corners increase, the redundant CFRTPCFs quickly accumulate inside the PTFE tube to block the nozzle. So the errors between the actual and ideal paths cause printing failure.

When analyzing the errors between the ideal and actual tracks, we first find that there are the obvious track errors at the corners, and the errors of different corner angles are different, as shown in Figure 4. The printing tracks transition with arc-like curves at the corners, and the gap areas enclosed between the ideal and actual tracks appear. The red lines in the figure represent the actual tracks and the green lines represent the ideal tracks. The larger the corner angles of the ideal tracks, the smaller the errors and the gap areas. We use the gap area to measure the track error and define it as the void area, as shown in Figure 5. The smaller the void area, the closer the actual track is to the ideal track.

In addition, experiments found that each approximate transition arc has a minimum radius, and if the curvature radius of the printing tracks at the corner is smaller than this minimum radius, printing will fail. The void area and the minimum radius of curvature were obtained by image processing. There were five samples in each group, and the average value of the five samples was taken as the result.

At the turning corner, continuous fiber tracks at different angles have different error areas, as shown in Figure 6a. It can be seen from the figure that with the increase of the laying angle at the turning corner, the void area decreases and tends to be stable, and the minimum curvature radius *R*_min_ at the corner gradually increases, as shown in Figure 6b.

When further exploring the source of the error between the ideal and actual track, it is found that this error is affected by the non-perpendicular fiber printing. Generally, in order to ensure the smooth extrusion of the CFRTPCF/PLA from the nozzle, the diameter of the nozzle must be larger than the cross-sectional diameter of the filament, as shown in Figure 7a, and due to the gap between the nozzle and the platform, as well as the pulling force on the filament generated by the movement of the nozzle, the filament forms a certain angle with the platform, and the center of the filament cross section does not coincide with the center of the nozzle, as shown in Figure 7c. When the nozzle moves in a long straight line, filament will always stick to the sidewall in the opposite direction of the nozzle movement. When the nozzle moves to the turning point, CGFRF/PLA will gradually follow the center of the nozzle, and finally stick to the sidewall in the opposite direction of the nozzle movement, as shown in Figure 7d. It can be clearly seen that there is an error between the nozzle path and the actual path of the CFRTPCF/PLA at the turning point causing excess fibers to accumulate in the PTFE tube inside the nozzle and resulting in printing failure.

In addition, when analyzing the errors between the ideal and actual track, it is also found that the track deviation is affected by the pulling force of the matrix on the filaments. Theoretically, even if the CFRTPCF track is deviated, its track should fall within the circular sweeping areas of the moving nozzle hole that constrain the continuous filaments. However, as shown in Figure 8, in the actual 30° corner it is found that the distance between the actual fiber corner and the ideal corner point is greater than the radius of the nozzle hole, which means that the actual track does not fall within the nozzle hole radius. While the red line segment represents the ideal track, the blue circle represents the nozzle hole, and the white represents the actual fiber track.

The continuous fiber printing track exceeds the nozzle hole area, which means that in the actual printing process, after the continuous fiber is melted and adhered to the surface of the substrate, the laid fiber track will be offset by the external force exceeding the nozzle inner hole area. This is because when the nozzle moves to the corner, the movement direction changes, and the sidewall of the nozzle exerts the pulling force along the new movement direction on the continuous fibers that are bonded to the surface of the substrate. When this pulling force is greater than the bonding force, the already bonded fibers shift at this pulling force, as shown in Figure 9. As the nozzle continues to move, the offset fibers adhere to new locations on the substrate surface in a continuous transitional manner, causing the original track to exceed the nozzle hole sweeping area and form a new curve track.

### 3.2. Model Establishment

In order to quantitatively analyze the deviation between the ideal and actual laying track, it is necessary to obtain a model of the actual fiber track according to the planning track.

#### 3.2.1. Line-Following Model

In order to build a model of the actual printing track of continuous fibers, the CFRTPCF track is segmented into small lines made of the fine points on the track and bonded to the surface of the substrate during laying. As shown in Figure 10, the motion model of the actual fiber printing track is explored in the 2D plane space.

Because the fiber filament is continuous, the actually laid fiber track is a smooth, first-order continuous curve. Assuming the diameter of the nozzle as D and the diameter of the fiber as d, when the nozzle moves linearly, the fiber track is consistent with the direction of the movement of the nozzle, and the fiber position is in the opposite direction of the nozzle movement relative to the nozzle center. ($x_{0},y_{0})$ and $\left(x_{f0},y_{f0}\right)$ are the coordinates of the nozzle center and the fiber section center at time *t*_0_, respectively, $\left(x_{1},y_{1}\right)$ and $\left(x_{f1},y_{f1}\right)$ are the coordinates of the corresponding center point at time $t_{1}$. The vector $L_{1}$ is obtained by connecting the point $\left(x_{f0},y_{f0}\right)$ and the point $\left(x_{1},y_{1}\right)$. The coordinate $\left(x_{f1},y_{f1}\right)$ of the next point of the fiber is on the line $L_{1}$ and is constrained to the inner circle of the nozzle. The schematic diagram of the model is shown in Figure 11a. From time $t_{0}$ to $t_{1}$, when $\left(x_{f0},y_{f0}\right)$ is inside the fiber nozzle, the fiber coordinates do not change, as shown in Figure 11b. The mathematical model of the above motion track of the fiber is established as shown in Equation (1). It can be seen from the analysis that the position of the current center point of the fiber is always on the connection line between the previous fiber center point and the current nozzle center point, and is within or tangent to the inner circle of the nozzle. It seems that the fiber position always follows the movement of the nozzle center, so this mathematical model is called as a line-following model.
(1)$$\left(x_{f1},y_{f1}\right)=\{\begin{array}{cc} \left(x_{f0},y_{f0}\right) & l\leq \frac{D-d}{2}\\ \left(x_{1},y_{1}\right)-\frac{\vec{L_{1}}}{\left|L_{1}\right|}\cdot \left(R_{n}-r_{f}\right) & other \end{array}$$

When the fiber-reinforced composite filament moves from a straight line into a turn, the position of the fiber filament relative to the nozzle center will change until the turning is completed, and the fiber is finally in the opposite direction of the linear motion of the nozzle after the turn. Figure 12a–d, respectively, indicates that the fiber-reinforced composite filament moves in a straight line from the initial state of $t_{0}$ to time $t_{1}$, turns from time $t_{1}$ to time $t_{2}$, and enters the turning state to the end of turn $t_{3}$, and also shows the positions of the fiber filament in the nozzle changes. Figure 13 shows the change of the position of the filament at different times in the nozzle at different turning angles. Turning track 1 represents an obtuse-angle turn, turning track 2 represents a right-angle turn, and turning track 2 represents an acute-angle turn.

For the track at the corner, the line-following model of Equation (1) can still be used for simulation, and the next point of the fiber track is on the connection line $L_{1}$. The comparisons between the simulation and the actual results at different angles are shown in Figure 14 and Figure 15.

From the above simulation results, it can be observed that the void area formed by the ideal track and the actual track is reduced to a certain extent, indicating that the track in the line-following model is more consistent with the actual printing track. Although the track of the line-following model is closer to the actual printing track, the void area is still large.

#### 3.2.2. Modified Line-Following Model

In the line-following model, it is found that the curvature radius of the model track at the corner is less than *R*_min_ (such as when the corner is 30°), and the first derivative of the track is discontinuous, which cannot meet the requirement of the fiber track. In order to solve this problem, a further hypothesis is put forward: when the curvature radius of the laid fiber track made of points is less than the minimum curvature radius *R*_min_ required by the material, the points are removed until the track minimum curvature requirements are met, and then the coordinates of the removed points are replaced by a transition curve with the first-order continuity shown in Figure 16.

The approximate calculation method of the curvature radius for the fiber track is shown in Figure 17. The continuous points $\left(x_{f-n},y_{f-n}\right)\to \left(x_{fn},y_{fn}\right)$ around the current fiber point $\left(x_{f0},y_{f0}\right)$ are fitted. In order to reduce the fitting error, an eighth-order polynomial is applied for calculation. The calculation formula is shown in Equation (2):
(2)$$\begin{array}{l} f=p_{1}x^{8}+p_{2}x^{7}+p_{3}x^{6}+p_{4}x^{5}+p_{5}x^{4}+p_{6}x^{3}+p_{7}x^{2}+p_{8}x^{1}+p_{9}\\ r_{x_{f0}}=\frac{{\left(1+{f^{\prime }}_{x_{f0}}{}^{2}\right)}^{\frac{3}{2}}}{\left|{f^{\prime \prime }}_{x_{f0}}\right|} \end{array}$$

When the curvature radius of the fiber track *r_xf0_* is greater than *R*_min_, the calculation is performed according to the line-following model. When it is less than *R*_min_, the coordinates of the previous fiber and nozzle points are removed until the minimum curvature is met, and the removed points are bridged by a transition curve. According to the above algorithm, the modified line-following model without the minimum curvature points is obtained:(3)$$\left(x_{f1},y_{f1}\right)=\{\begin{array}{ll} (x_{f0},y_{f0}), & l<\frac{D-d}{2}\\ \left\{ \begin{array}{ll} \left(x_{1},y_{1}\right)+\frac{\vec{L_{1}}}{\left|L_{1}\right|}\cdot \left(R_{n}-r_{f}\right)\cdot \vec{v_{1}}, & r_{x_{f0}}>R_{\min}\\ \;\mathrm{null}\;\left(\mathrm{remove}\left(x_{0},y_{0}\right),\left(x_{f0},y_{f0}\right)\right), & r_{x_{f0}}\leq R_{\min} \end{array},\right. & l\geq \frac{D-d}{2} \end{array}$$

The model simulation results are shown in Figure 18. The orange curves in the figure are the actual curves, and the blue broken lines are the line-following model after removing the minimum curvature point. The removed points are bridged with a transition curve. As shown in Figure 19, the bridge transition curve needs to meet the following two requirements:(1)The transition curve is tangent to the fiber track at the start and end points.(2)The minimum curvature of the transition curve is greater than or equal to *R*_min_.

According to the requirements of the bridge transition curve, the Bezier curve is selected as the bridge curve, which uses the starting point, the end point, and a certain number of control points to control a smooth transition curve. According to the number of control points, Bezier curves are divided into first-order Bezier curves (0 control points), second-order Bezier curves (1 control point), third-order Bezier curves (2 control points), and so on. The Bezier curve always passes through the start and end points, and is always tangent to the feature polygon at the start and end points. In the transition scheme, we apply the cubic (third-order) Bezier with known start and end points, and the start and end straight lines passing through the two points are also known. Therefore, two control points can be found on the start and end straight lines.

One smooth cubic Bezier curve can be controlled through four points $\left[P_{0},P_{1},P_{2},P_{3}\right]$. The smooth curve is tangent to line $P_{0}P_{1}$ at point $P_{0}$ and line $P_{2}P_{3}$ at point $P_{3}$. The above two tangent conditions can satisfy the continuity of the transition curve. The minimum curvature requirement is satisfied by controlling the positions of the control points $P_{1}$ and $P_{2}$. The known control points $P_{0}$ and $P_{3}$ are the starting point $\left(X_{f0},Y_{f0}\right)$ and the ending point $\left(X_{f1},Y_{f1}\right)$ of the known track, the coordinates of the previous point of $\left(X_{f0},Y_{f0}\right)$ is $\left(X_{f},Y_{f}\right)$, the next point of $\left(X_{f1},Y_{f1}\right)$ is $\left(X_{f2},Y_{f2}\right)$, and the unknown control points are $P_{1}$ and $P_{2}$. Connecting point $\left(X_{f},Y_{f}\right)$ and point $\left(X_{f0},Y_{f0}\right)$ results in a straight line $L_{6}$, connecting point $\left(X_{f1},Y_{f1}\right)$ and point $\left(X_{f2},Y_{f2}\right)$ results in a straight line $L_{7}$, and $L_{6}$ and $L_{7}$ intersect at point $P_{5}$. The control point $P_{1}$ is on the straight line $L_{6}$, the control point $P_{2}$ is on the straight line $L_{7}$, and the $K_{p}$ scale factor is used to describe the positions of the control points $P_{1}$ and $P_{2}$, as shown in Figure 20b.

The bridging curve adopts the cubic Bezier curve parametric equation as follows:(4)$$X\left(t\right)=x_{0}\cdot {\left(1-t\right)}^{3}+x_{1}\cdot {\left(1-t\right)}^{2}\cdot t+x_{2}\cdot \left(1-t\right)\cdot t^{2}+x_{3}\cdot t^{3}\;Y\left(t\right)=y_{0}\cdot {\left(1-t\right)}^{3}+y_{1}\cdot {\left(1-t\right)}^{2}\cdot t+y_{2}\cdot \left(1-t\right)\cdot t^{2}+y_{3}\cdot t^{3}\;P_{1}=P_{0}+\frac{{\vec{L}}_{6}}{\left||{\vec{L}}_{6}|\right|}\cdot K_{p}\cdot ||{\vec{L}}_{6}||,P_{2}=P_{3}+\frac{{\vec{L}}_{7}}{\left||{\vec{L}}_{7}|\right|}\cdot K_{p}\cdot ||{\vec{L}}_{7}||\;P_{0}=\left(x_{0},y_{0}\right),P_{1}=\left(x_{1},y_{1}\right),P_{2}=\left(x_{2},y_{2}\right),P_{3}=\left(x_{3},y_{3}\right)\;K_{p},t\in \left(0,1\right)$$

Derivative of the above formula can be obtained:(5)$$\left\{ \begin{array}{c} X^{\prime }\left(t\right)=3\cdot \left(x_{3}-3x_{2}+3x_{1}-x_{0}\right)\cdot t^{2}+6\cdot \left(x_{2}-2x_{1}+x_{0}\right)\cdot t+3\cdot \left(x_{1}-x_{0}\right)\\ Y^{\prime }\left(t\right)=3\cdot \left(y_{3}-3y_{2}+3y_{1}-y_{0}\right)\cdot t^{2}+6\cdot \left(y_{2}-2y_{1}+y_{0}\right)\cdot t+3\cdot \left(y_{1}-y_{0}\right)\\ X^{\prime \prime }\left(t\right)=6\cdot \left(x_{3}-3x_{2}+3x_{1}-x_{0}\right)\cdot t+6\cdot \left(x_{2}-2x_{1}+x_{0}\right)\\ Y^{\prime \prime }\left(t\right)=6\cdot \left(y_{3}-3y_{2}+3y_{1}-y_{0}\right)\cdot t+6\cdot \left(y_{2}-2y_{1}+y_{0}\right) \end{array}\right.$$

Then the radius of curvature at any point of the bridge curve is:(6)$$r\left(t\right)=\frac{{\left[{X^{\prime }}^{3}\left(t\right)+{Y^{\prime }}^{2}\left(t\right)\right]}^{\frac{3}{2}}}{X^{\prime }\left(t\right)Y^{\prime \prime }\left(t\right)-X^{\prime \prime }\left(t\right)Y^{\prime }\left(t\right)}=r\left(t,K_{p}\right)$$

When the radius of curvature of the bridge curve is the minimum value, it satisfies:(7)$$\frac{dr\left(t,K_{p}\right)}{dt}=0$$

We come to:(8)$$t=t\left(K_{p}\right)$$

When the minimum curvature radius *r*(*t*,*K_p_*) of the bridge curve is equal to *R*_min_, the *K_p_* value is obtained:(9)$$r\left(t,K_{p}\right)=r\left(t\left(K_{p}\right),K_{p}\right)=R_{\min}$$

Then, we substitute *K_p_* into Equation (4) to obtain all points $\left(X_{b},Y_{b}\right)$ of the bridge curve, as shown in Equation (10):(10)$$\left(X_{b},Y_{b}\right)=\left(X\left(t\right),Y\left(t\right)\right)$$

Finally, the modified line-following model without the minimum curvature point is obtained, as shown in Equation (11):(11)$$\left(x_{f1},y_{f1}\right)=\{\begin{array}{ll} (x_{f0},y_{f0}), & l<\frac{D-d}{2}\\ \left\{ \begin{array}{cc} \left(x_{1},y_{1}\right)+(R-r)\cdot \frac{\vec{L_{1}}}{\left||\vec{L_{1}}|\right|}, & r_{x_{f0}}>R_{\min}\\ \left(X_{b},Y_{b}\right), & \;r_{x_{f0}}\leq R_{\min} \end{array},\right. & l\geq \frac{D-d}{2} \end{array}$$

## 4. Results and Discussion

The modified line-following model without the minimum curvature point is simulated. The minimum curvature radius *R*_min_ of the corner is obtained in Figure 6b, and the results of the variation of the variable *K_P_* and the void areas with the angles are obtained. The results of the minimum curvature modified line-following model for multi-angle corner tracks are shown in Figure 21 below.

From Figure 22, it can be concluded that the void area of the modified line-following model is greatly reduced when compared with the line-following model by removing the small curvature point and tracks obtained from the modified model, which are consistent with the actual printing track. With further observation of the variation trend of the void areas with the angles, it is seen that when the turning angle is below 60°, the void areas are within about 0.2 mm^2^, and when the turning angle is above 60°, the void areas are controlled within 1 mm^2^, which indicates the applicability of the model to small angle corners. As the turning angle increases, the errors of the modified line-following model and the line-following model gradually tend to be consistent and the correction effect of the modified line-following model gradually disappears at large corners (obtuse angles), which means that the points where the radius of curvature of the fiber track is smaller than *R*_min_ is gradually decreasing at the large corner, which further illustrates the correction effect of the modified model.

The parameter *K_p_* varies with the corner angle as shown in Figure 23. The parameter *K_p_* and the angle show a relatively stable linear change, and the value of the parameter *K_p_* under other turning corners can be predicted by interpolation. The above results verify the accuracy, rationality, and stability of the modified line-following model after removing the small curvature points.

## 5. Summary

During the actual CFRTPC 3D printing, the problem of nozzle clogging and printing interruption often occurs. Aiming at this problem, this paper analyzes the difference between the theoretical planning and the actual printing CFRTPC track, and finds that their tracks do not overlap, and the actual printing track is shorter than the theoretical planning track. When CFRTPC is printed, the system feeds the material according to the theoretical track, which causes the excess CFRTPC material to be blocked at the internal throat of the nozzle. Based on the analysis of the above problems, this paper theoretically explores the primary causes of the printing fiber track errors. The specific conclusions are as follows.

(1)In this paper, the closed area of the actual printing track and the theoretical planning track is used to characterize the trajectory error. The void area and the minimum radius of curvature were measured by image processing.(2)Because the diameter of the inner hole of the CFRTPC printing nozzle is larger than the diameter of the CFRTPC, a line-following model is proposed. Compared with the actual printed results, the simulation results show that the void area formed by the ideal track and the actual track is reduced to a certain extent. Although the track of the line-following model is closer to the actual printing track, the void area is still large. Furthermore, the smaller the corner angle, the greater the error.(3)When the line-following model cannot accurately reflect the actual printing model at small-angle turning corners, a modified line-following model is proposed. This model better simulates the actual fiber laying track corresponding to the theoretical planning track at different corner angles. Results show that when the turning angle increases, the errors of the modified line-following model and the line-following model gradually become consistent, and the correction effect of the modified line-following model gradually disappears at large corners (obtuse angles), which means that the points where the radius of curvature of the fiber track is smaller than *R*_min_ is gradually decreasing at the large corner, which further illustrates the correction effect of the modified model.

The research in this paper can provide a theoretical basis for the trajectory planning of continuous fiber. In this way, the failure of a blocked nozzle in the printing process is lessened, and the printing reliability of CFRTPC is improved. Further improvement of the printing trajectory accuracy, improvement of printing accuracy, less porosity, and improvement of the quality of continuous fiber-reinforced composites is needed.

## Figures and Tables

**Figure 1 polymers-14-02730-f001:**
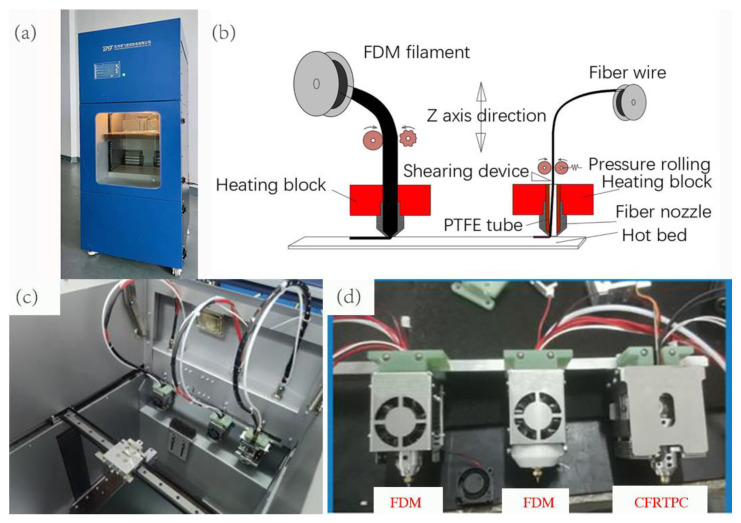
CFRTPC printer. (**a**) Printer appearance. (**b**) Extrusion principle diagram of two printing nozzles. (**c**) Printing platform. (**d**) Stand-alone printing head.

**Figure 2 polymers-14-02730-f002:**
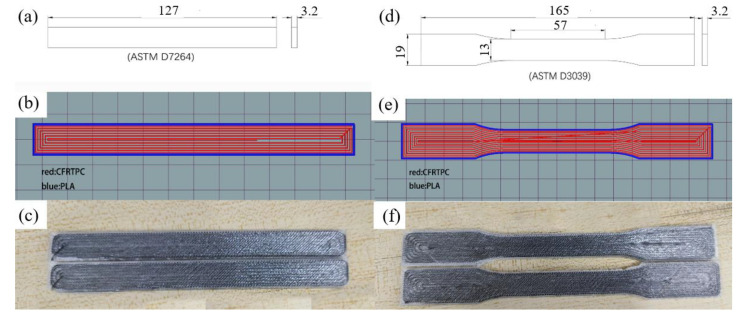
Standard sample, (**a**) size of standard three-point bending specimen, (**b**) intra-layer tracks of bending specimen, (**c**) the printed real bending specimens, (**d**) size of standard tensile specimen, (**e**) intra-layer tracks of tensile specimen, and (**f**) the printed real tensile specimens.

**Figure 3 polymers-14-02730-f003:**
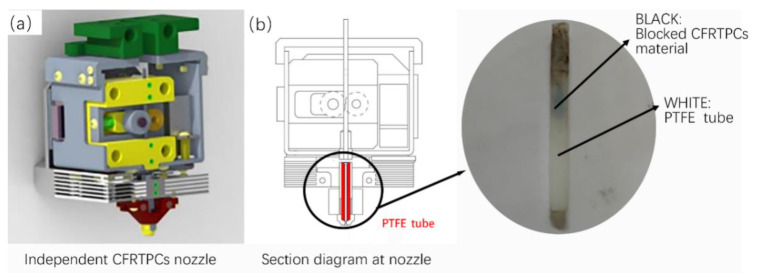
CFRTPC nozzle structure and blockage, (**a**) 3D image of a CFRTPC independent printing nozzle, (**b**) schematic diagram of the position of PTFE tube in nozzle.

**Figure 4 polymers-14-02730-f004:**
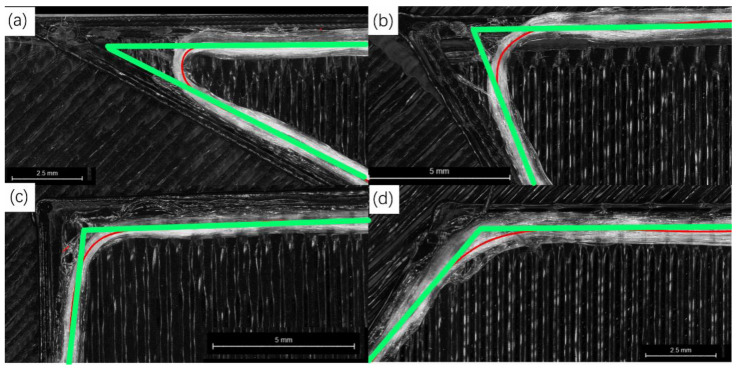
Printing with different corner angles, (**a**) 30°, (**b**) 60°, (**c**) 90°, (**d**) 120°. White: CGFRF/PLA. Black: PLA. Red: extracted CGFRF trace lines. Green: ideal tracks.

**Figure 5 polymers-14-02730-f005:**
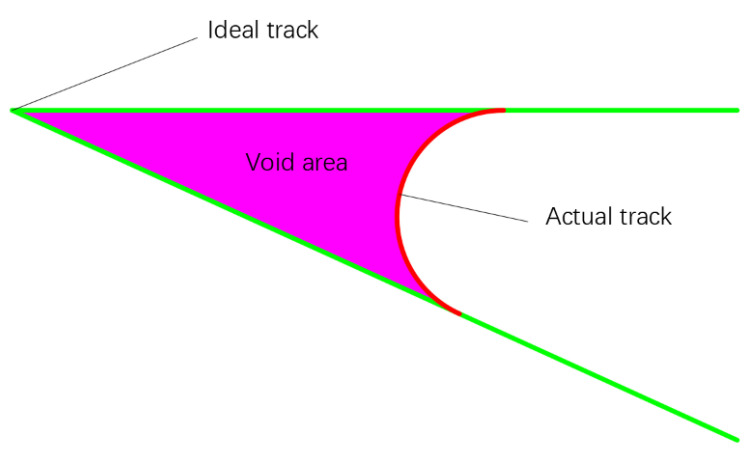
Schematic diagram of void area.

**Figure 6 polymers-14-02730-f006:**
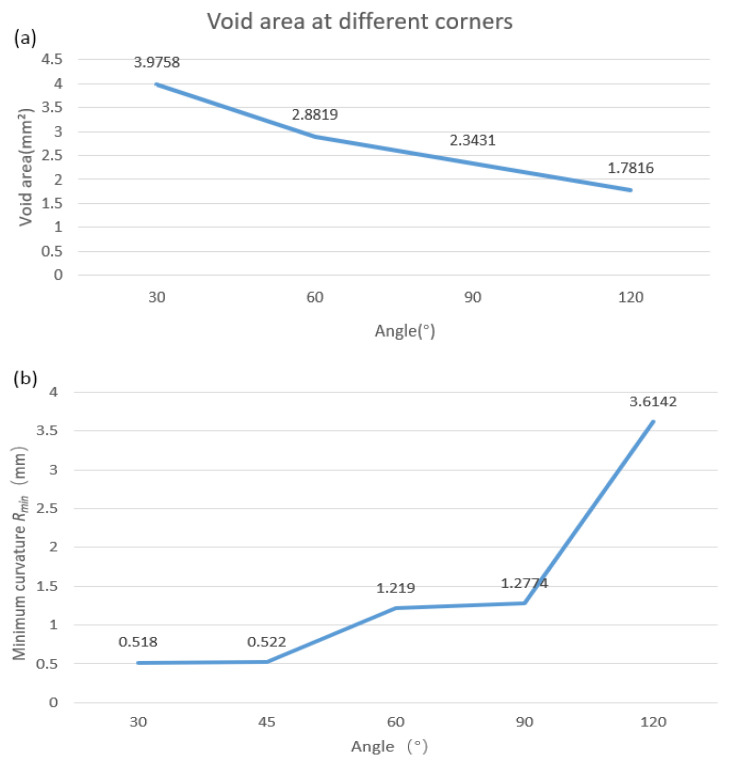
Actual track result, (**a**) void area change curve with angle, (**b**) schematic diagram of the minimum curvature *R*_min_ at the corner changing with the angle.

**Figure 7 polymers-14-02730-f007:**
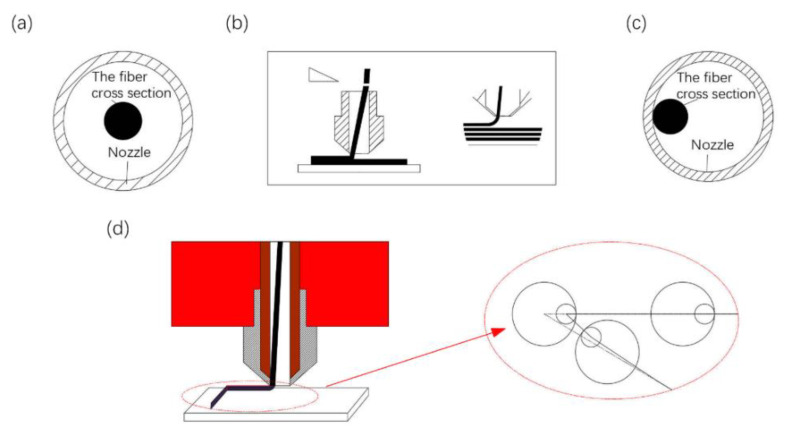
Fiber pose diagram in the nozzle, (**a**) initial position of fiber, (**b**) fiber upright state, (**c**) position during fiber movement, (**d**) print track graph.

**Figure 8 polymers-14-02730-f008:**
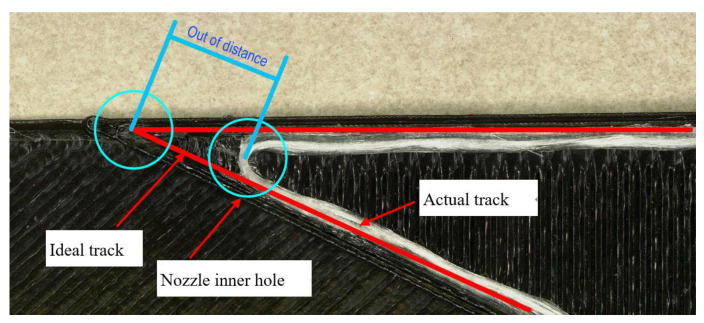
Ideal track and actual track.

**Figure 9 polymers-14-02730-f009:**
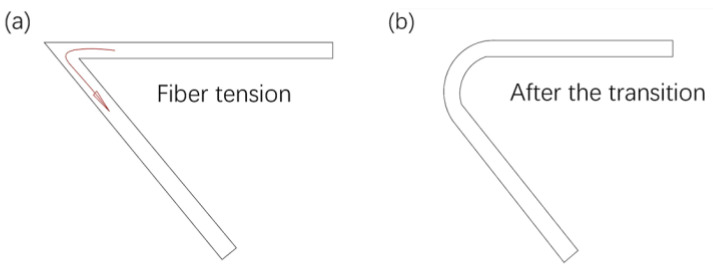
Actually print the transition curve after offset, (**a**) original track, (**b**) after the transition of external force.

**Figure 10 polymers-14-02730-f010:**
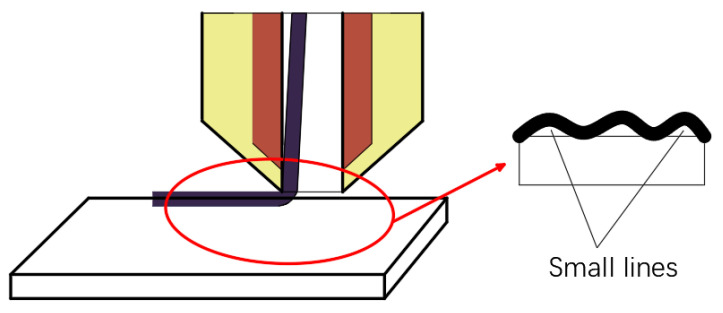
Bonding model.

**Figure 11 polymers-14-02730-f011:**
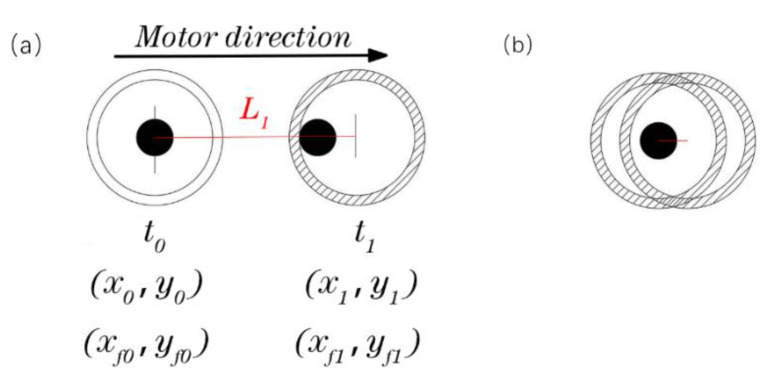
Line-following model, (**a**) line-following, (**b**) special cases.

**Figure 12 polymers-14-02730-f012:**
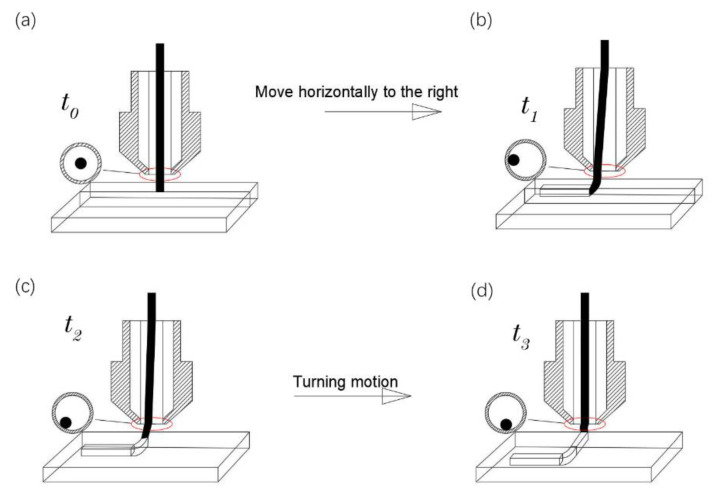
Corner motion model, (**a**) initial printing state, (**b**) straight line printing status, (**c**) initial turning state, (**d**) turning completion state.

**Figure 13 polymers-14-02730-f013:**
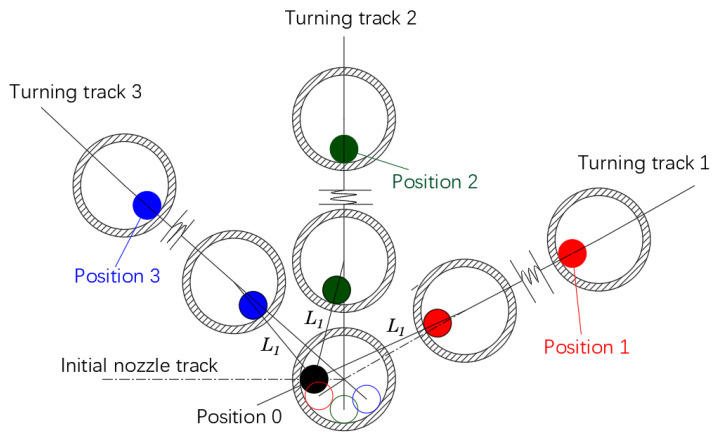
Schematic diagram of following at different corner angles.

**Figure 14 polymers-14-02730-f014:**
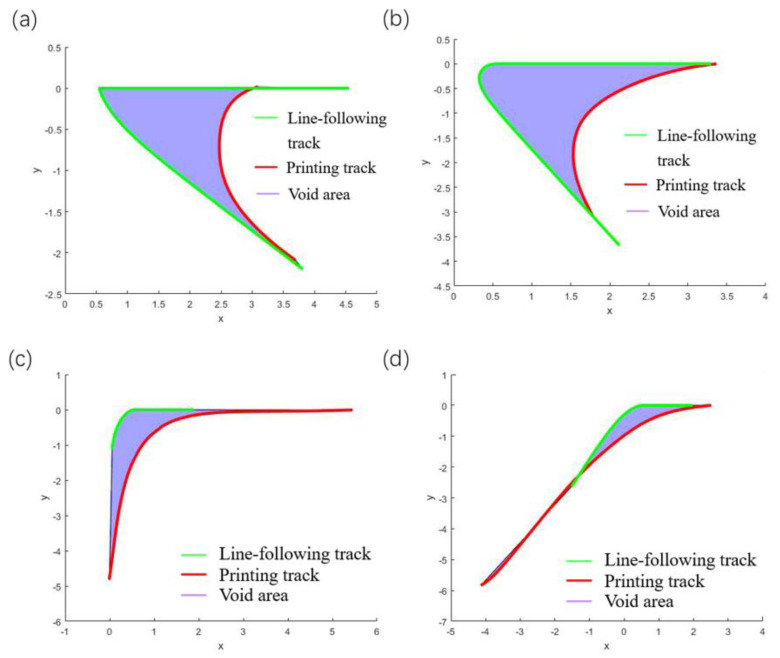
Comparison of simulation results of straight lines-following corners from different angles, (**a**) 30°, (**b**) 60°, (**c**) 90°, (**d**) 120°.

**Figure 15 polymers-14-02730-f015:**
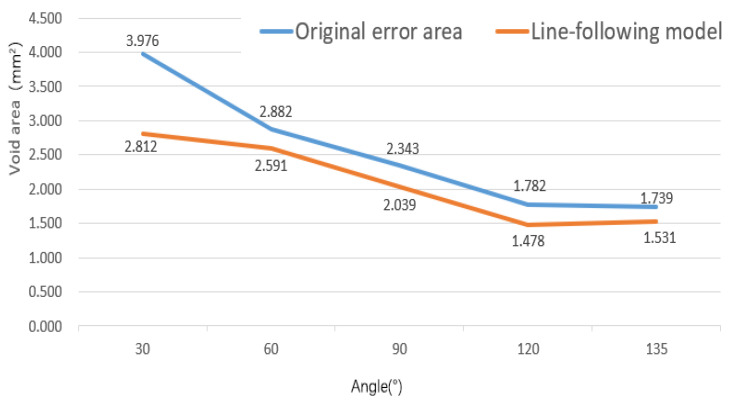
Void areas change with angles.

**Figure 16 polymers-14-02730-f016:**
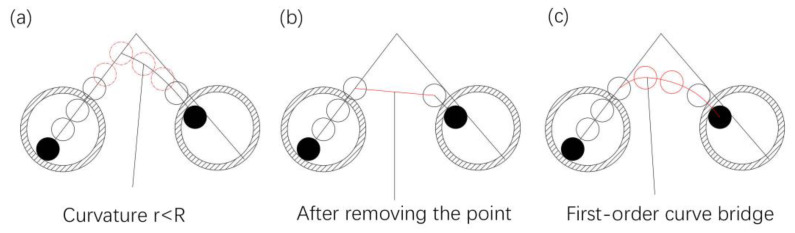
The principle of removing points at the corner. (**a**) Track points that make up the curve whose curvature radius is less than *R*_min_, (**b**) track after removing the points, causing the track curvature radius to be less than *R*_min_, (**c**) first-order transition curve replaces the removed the points.

**Figure 17 polymers-14-02730-f017:**
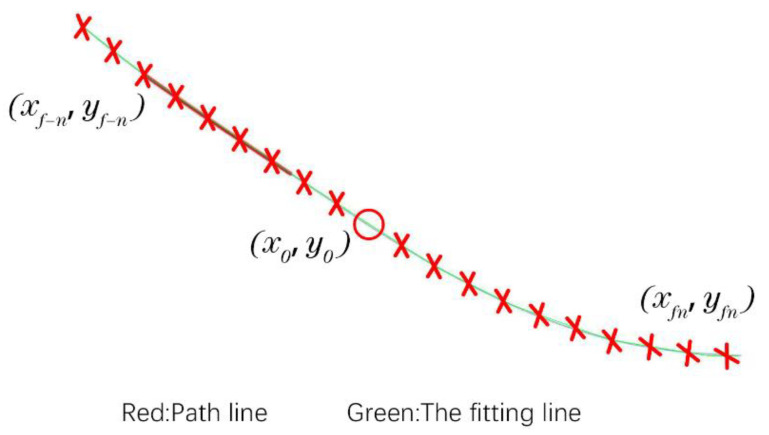
Approximate calculation of curvature radius for laying fiber track point.

**Figure 18 polymers-14-02730-f018:**
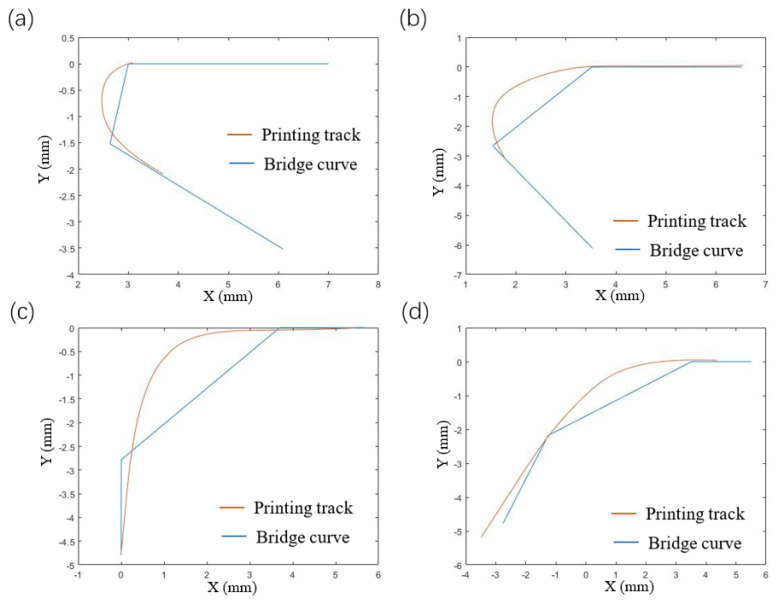
Simulation of removing minimum curvature points at different corners, (**a**) 30°, (**b**) 60°, (**c**) 90°, (**d**) 120°.

**Figure 19 polymers-14-02730-f019:**
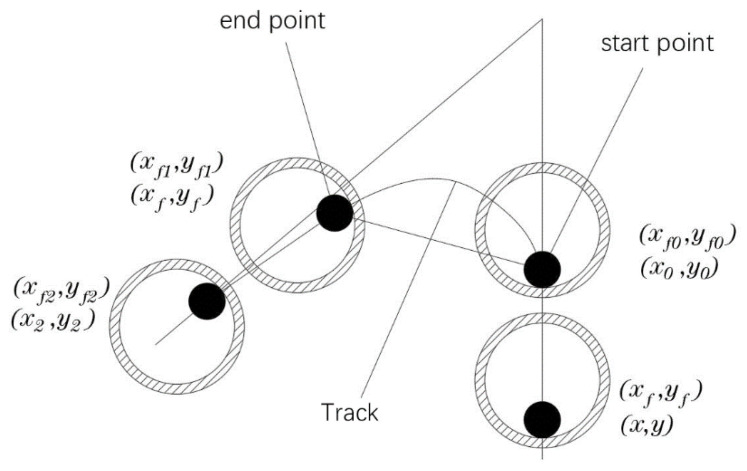
Transition connection diagram.

**Figure 20 polymers-14-02730-f020:**
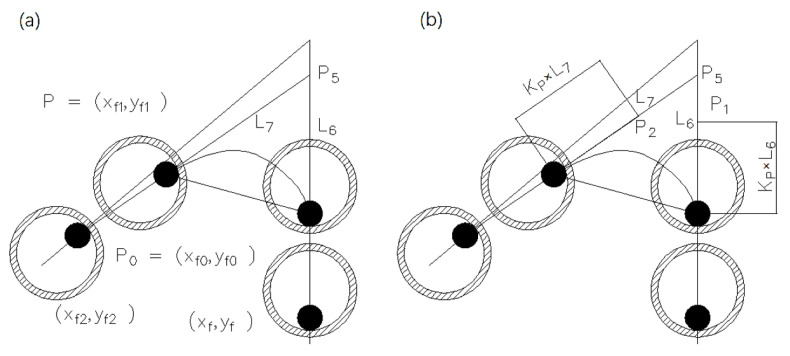
Schematic diagram of bridging the transition curve, (**a**) control line at bridge, (**b**) schematic diagram of variable control points.

**Figure 21 polymers-14-02730-f021:**
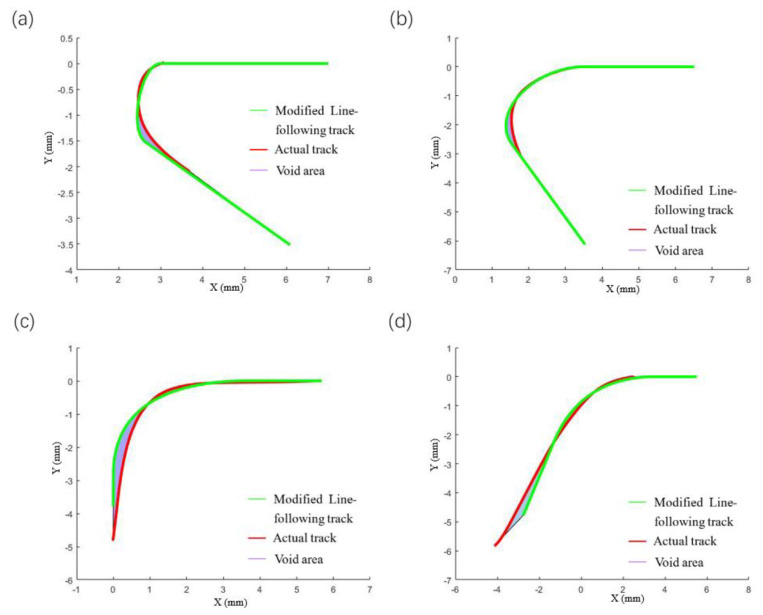
Test of the smallest area of bridge after removing the smallest curvature at different corner angles, (**a**) 30°, (**b**) 60°, (**c**) 90°, (**d**) 120°.

**Figure 22 polymers-14-02730-f022:**
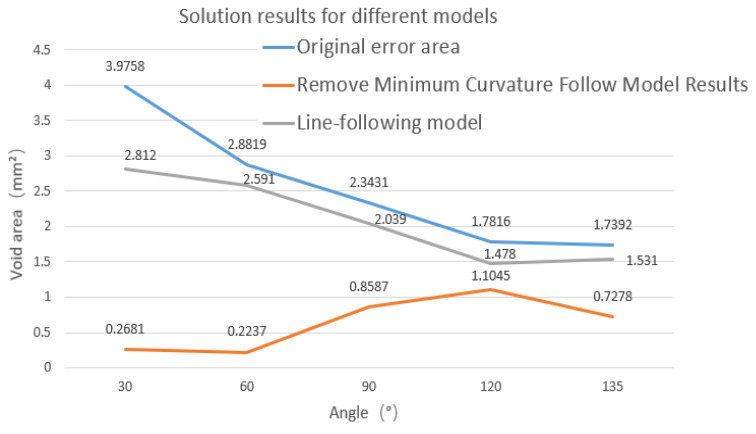
Simulation parameters at different angles.

**Figure 23 polymers-14-02730-f023:**
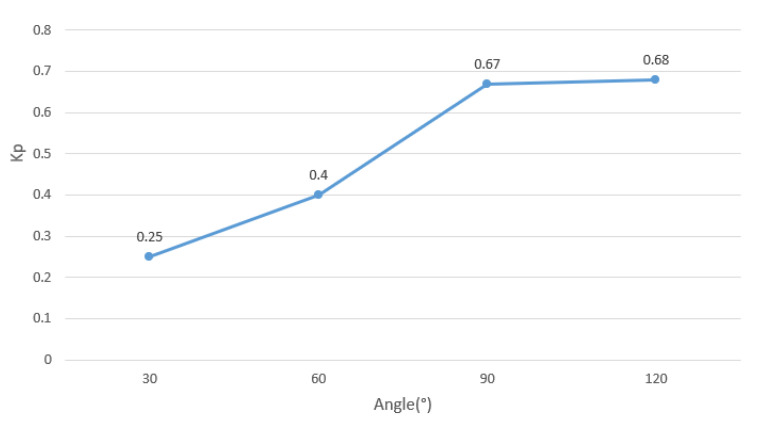
*K_p_* results as a function of corner angle.

**Table 1 polymers-14-02730-t001:** PLA material parameters [35].

Property	Value
Filament diameter, mm	1.75
Density, g/cm^3^	1.2
Tensile strength, MPa	72
Bending strength, MPa	90
Printing temperature, °C	190–230
Base plate temperature, °C	45–60

**Table 2 polymers-14-02730-t002:** CFRTPCFs material parameters [36].

Property	Value
Filament diameter, mm	0.40
Surface roughness, um	18
Degree of bending, %	25.8
Tensile strength, MPa	375 MPa
Twist or not	Twist
Twist direction	S

**Table 3 polymers-14-02730-t003:** Printing parameters.

Material	Parameter Description
Basic material	Polylactic acid (PLA), provided by eSUN
Continuous fiber material	Self-made CGFRF/PLA in the laboratory
Substrate printing temperature, °C	210
Continuous fiber material printing temperature, °C	200
Continuous fiber material printing speed, mm/min	400
Fiber printing layer height, mm	0.25
Base nozzle diameter, mm	0.4
Fiber nozzle diameter r, mm	1.2
Ambient temperature	25 °C

## Data Availability

The data are contained within the article.

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
