# Peer review of "Research on the Simulation Model of Continuous Fiber-Reinforced Composites Printing Track"

_polymers, 2022, doi:10.3390/polym14132730_

Round 1
Reviewer 1 Report
In general, the article is interesting and presents innovative results. Here are just a few suggestions for improvement:
1) Item "2.13. D printer" or "2.1" ?
2) Please explain how the results in Table 1 were determined.
3) In Table 1, the property "Tensile strength, MPa" appears twice, with a different value.
4) Conclusion is better than summary.
Author Response
1) Item "2.13. D printer" or "2.1" ?
Response 1: I'm sorry that my expression confused you. I have corrected it in the article. (line197-198)
2) Please explain how the results in Table 1 were determined.
Response 2: Thank you very much. The PLA material parameters in Table 1 are provided by PLA supplier eSUN (https://www.esun3d.net/). Its references have been added to the revised manuscript. (line197 and 555)
3) In Table 1, the property "Tensile strength, MPa" appears twice, with a different value.
Response 3: Because of my carelessness, I filled the tensile strength(370MPa) in Table 2 into Table 1. The incorrect data in Table 1 has been deleted. (line197-198)
4) Conclusion is better than summary.
Response 4: Thanks for your advice. The summary has been revised in the revised manuscript. (line457-488)
Please see the attachment.

Reviewer 2 Report
Comments
This paper studied fiber reinforced composites printing track. The outcome of the paper is interesting however, there are several aspects that need to be improved. The reviewer can only recommend for publication if the author satisfactorily address the following major comments in the revised version.
1. The research gap from the literature review should be clearly presented.
2. The research questions and justification of selecting variables should be highlighted.
3. Which test standards was considered in this study? How many replicate samples were tested in each category?
5. The novelty of the study should be highlighted more clearly at the end of introduction section. How this study is different from the published study in literature?
6. How the outcome of this study will benefit researchers and end users? This need to be highlighted in introduction or end of conclusion.
7. The 3D printed structures is interesting but not novel. Therefore, the recent application in this area should be discussed in introduction section to improve the background study. Recently, the application, performance and challenges of 3D printed materials are highlighted in [Ref: 3D-printed concrete: Applications, performance, and challenges], and the continuous fibre composites has attracted attention in structural application [Ref: Behaviour of continuous fibre composite sandwich core under low-velocity impact]. Suggest to include them in introduction section with proper citations to improve the background study.
I would be happy to see the revised version to understand how these comments are being addressed.
Author Response
- The research gap from the literature review should be clearly presented.
Response 1: Thanks for your advice. Please check our supplement to the introduction according to your suggestion.(line144-152)
- The research questions and justification of selecting variables should be highlighted.
Response 2: Thanks. The main problem of this paper is that in the printing process of continuous fiber reinforced composites, the theoretical trajectory of fiber does not coincide with the actual trajectory, which will cause the failure of printing nozzle blockage. The main problem of this paper is that in the printing process of continuous fiber reinforced composites, the theoretical trajectory of fiber does not coincide with the actual trajectory, which will cause the failure of printing nozzle blockage. This problem is caused by the inconsistency between the print nozzle and the fiber diameter directly, and the fiber's inherent radius of curvature. In order to explore the mechanism of printing trajectory error of fiber, this paper develops a mathematical model of continuous fiber printing by using nozzle structure and fiber minimum radius of curvature as research variables. (line152-160)
- Which test standards was considered in this study? How many replicate samples were tested in each category?
Response 3: Thank you for your comments. The experimental part of this paper tests the error between the actual printing trajectory and the theoretical trajectory in the actual printing process. The measurement method is to obtain the void area and the minimum radius of curvature were obtained by image processing. There were 5 samples in each group, and the average value of the 5 samples was taken as the result. This part has been added to the revised manuscript. (line232-234)
- The novelty of the study should be highlighted more clearly at the end of introduction section. How this study is different from the published study in literature?
- How the outcome of this study will benefit researchers and end users? This need to be highlighted in introduction or end of conclusion.
Response 4 and 5: Thanks for your advice. The summary has been revised in the revised manuscript. (line144-171, line454-485)
- The 3D printed structures is interesting but not novel. Therefore, the recent application in this area should be discussed in introduction section to improve the background study. Recently, the application, performance and challenges of 3D printed materials are highlighted in [Ref: 3D-printed concrete: Applications, performance, and challenges], and the continuous fibre composites has attracted attention in structural application [Ref: Behaviour of continuous fibre composite sandwich core under low-velocity impact]. Suggest to include them in introduction section with proper citations to improve the background study.
Response 6: I'm really sorry that I didn't read the latest articles because of my negligence. I have read two articles carefully and got a lot of inspiration. The article is worthy and I have quoted it in the introduction. Thank you very much. (line42-44)
Please see the attachment.

Round 2
Reviewer 2 Report
I have no further comments